# Optimal Intervention on Weighted Networks via Edge Centrality

Dongyue Li
li.dongyu@northeastern.edu
Northeastern University
Boston, MA, USA

Tina Eliassi-Rad
t.eliassirad@northeastern.edu
Northeastern University
Boston, MA, USA

Hongyang R. Zhang
ho.zhang@northeastern.edu
Northeastern University
Boston, MA, USA

## ABSTRACT

We consider the problem of diffusion control via interventions that change network topologies. We study this problem for general weighted networks and present an iterative algorithm, Frank-Wolfe-EdgeCentrality, to reduce the spread of a diffusion process by shrinking the network's top singular values. Given an edge-weight reduction budget, our algorithm identifies the near-optimal edge-weight reduction strategy to minimize the sum of the largest $r$ eigenvalues of $W^\top W$, where $W$ is the network weight matrix. Our algorithm provably converges to the optimum at a rate of $O(t^{-1})$ after $t$ iterations; each iteration only requires a nearly-linear runtime in the number of edges.

We perform a detailed empirical study of our algorithm on a wide range of weighted networks. In particular, we apply our approach to reduce edge weights on mobility networks (between points of interest and census block groups), which have been used to model the spread of COVID-19. In SEIR model simulations, our algorithm reduces the number of infections by 25.70% more than existing approaches, averaged over three weighted graphs and eight mobility networks. Meanwhile, the largest singular value of the weight matrix $W$ decreases by 25.48% more than existing approaches on these networks. An extension of our algorithm to temporal mobility networks also shows an effective reduction in the number of infected nodes.

## KEYWORDS

Targeted Immunization, Graph Algorithms, Epidemic Spreading, Edge Centrality.

**ACM Reference Format:**
Dongyue Li, Tina Eliassi-Rad, and Hongyang R. Zhang. 2022. Optimal Intervention on Weighted Networks via Edge Centrality. In *epiDAMIK 2022: 5th epiDAMIK ACM SIGKDD International Workshop on Epidemiology meets Data Mining and Knowledge Discovery, August 15, 2022, Washington, DC, USA.* ACM, New York, NY, USA, 11 pages.

## 1 INTRODUCTION

Network diffusion processes such as disease spread and information dissemination are ubiquitous in our increasingly well-connected society. In many applications, one would like to control the outcome of network diffusion processes by "designing the network connections." For example, a classical problem in this domain is influence maximization (Chen et al., 2010, Kempe et al., 2003), where the goal is to identify a small set of nodes to maximize the spread of a diffusion process such as for the adoption of a new product (Singer, 2012). In other settings, such as disease control, the goal is

instead to design a network-based immunization strategy to slow down the disease spread (Ganesh et al., 2005, Wang et al., 2003).

The problem of diffusion control has gained recent interest in identifying non-pharmaceutical interventions such as lockdown to slow the spread of the SARS-CoV-2 virus. For example, Chang et al. (2021a) and Chang et al. (2021b) introduce mobility-based modeling to study the spread of the COVID-19 pandemic. Their approach consists of two major components. First, mobility networks that describe the movement of people from neighborhoods to points of interest are constructed based on mobility records. Second, a metapopulation Susceptible-Exposed-Infectious-Recovered (SEIR) model is overlaid on the mobility network. A major finding of Chang et al. (2021a) is that mobility network models can accurately fit the reported COVID-19 case counts. Based on this finding, they simulate various interventions such as capping the maximum occupancy of places to understand their effect on reducing infection.

We study network-based interventions to reduce the number of infected nodes in general weighted and directed networks. Suppose there is an epidemic spreading on the network. How can we slow down the spread of the epidemic in the network, subject to reducing the *edge weights* by a limited amount, due to budget constraints? For example, the weight of an edge from a census block group to a place in a mobility network represents the amount of traffic between them. Reducing the weight of this edge corresponds to mobility reduction.

The spreading rate of a diffusion process is closely related to the spectral properties of a network. An important result from the epidemics literature is that the epidemic threshold–below which a diffusion process will die out quickly–scales linearly with the largest (in module) eigenvalue (denoted as $\lambda_1$) of the adjacency matrix of the network (Chakrabarti et al., 2008). As Prakash et al. (2012) proved, this result generalizes to various epidemic models. Thus, a natural strategy for slowing down a diffusion process is to remove nodes or edges to reduce $\lambda_1$ of a network's adjacency matrix. However, minimizing $\lambda_1$ subject to removing a fixed number of nodes or edges is NP-hard via a reduction to the independent set problem (Chen et al., 2015, Karp, 1972). Therefore, various heuristics are proposed to solve this problem in practice. For example, Chen et al. (2015) show that by choosing nodes with the highest *node centrality* scores (i.e., a node's value in the eigenvector corresponding to the largest eigenvalue) in a greedy approach, one can reduce the largest eigenvalue and achieve notable reductions in the number of infected nodes. Tong et al. (2012) have likewise shown that choosing edges with the highest *edge centrality* scores (i.e., the product of the node centrality scores from both endpoints of an edge) in a greedy algorithm is a scalable and effective approach. Chen et al. (2018) further quantified the approximation ratio of these greedy approaches by using techniques from submodular optimization. In light of these works, one natural approach to solving

*epiDAMIK 2022, August 15, Washington DC*

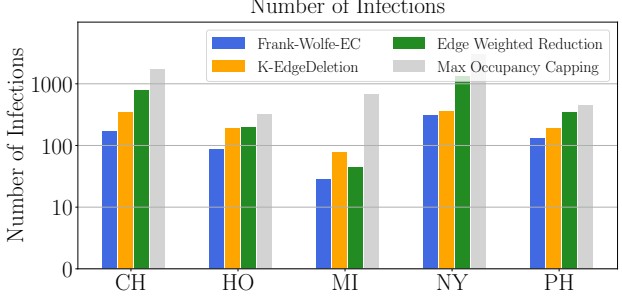

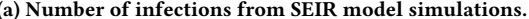

**(a) Number of infections from SEIR model simulations.**

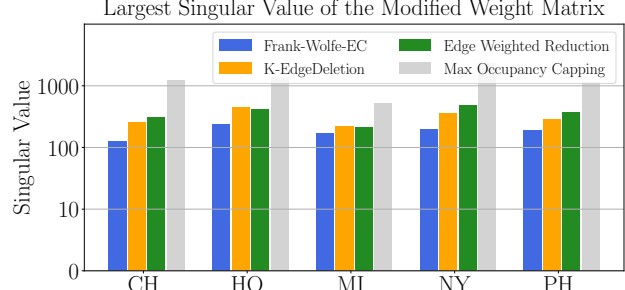

**(b) Largest singular value of modified network weight matrix.**

**Figure 1: Comparison of Frank-Wolfe-EC, K-EdgeSelection, edge weighted reduction, and max occupancy capping in different weighted mobility networks. Notice that Frank-Wolfe-EC reduces the number of infections and the largest singular value more significantly than previous approaches on all networks. For details, see Section 4.**

the intervention problem is minimizing the top eigenvalue(s) using edge-weight reduction, and the following questions arise. Does this approach perform well (e.g., compared to greedy algorithms)? Can this problem be solved efficiently in polynomial time? This work provides affirmative answers to these questions by developing an iterative algorithm that provably converges to the optimum of the minimization problem on weighted networks. See Figure 1 for an illustration of our results.

## 1.1 Our Contributions

We begin by showing that the edge centrality measures from Tong et al. (2012) are equal to the gradient of the largest eigenvalue $\lambda_1(W^\top W)$ (up to scaling), where $W$ is the network weight matrix. We validate that reducing the edge weights of the highest edge centrality scores effectively reduces the number of infected nodes in a weighted network.

Then, we develop a new algorithm, *Frank-Wolfe Edge Centrality minimization*, or Frank-Wolfe-EC, to minimize the sum of the largest $r$ eigenvalues of $W^\top W$, subject to an edge-weight reduction budget. At every iteration, our algorithm finds a descent direction that correlates the least with the edge centrality scores, given their gradient interpretation above. A naive approach to finding the descent direction requires solving a linear program (LP). Instead, we present a nearly-linear time algorithm (in the number of edges) by characterizing the LP's optimum as a greedy selection of edges with the highest edge centrality scores. Additionally, we prove that Frank-Wolfe-EC converges to the global minimum at a rate of $O(t^{-1})$ after $t$ iterations.

We evaluate our algorithm by simulating an epidemic model on publicly available weighted graphs and mobility networks. First, on three weighted graphs, our approach achieves on average **10.46%** improvement over baselines during SEIR model simulations. Meanwhile, the largest singular value decreases by an average of **11.42%** more than the baselines. Second, we apply our approach to reduce edge weights on mobility networks. Our algorithm reduces the infected populations by **30.17%** and the largest singular value by **30.75%** more than prior approaches on average. Finally, we extend our algorithm to tackle temporal networks by allocating the weight-reduction budget proportionally to a network's largest

singular value. We find that on sequences of temporal mobility networks, this strategy reduces infections by **39.82%** more than other heuristics.

**Summary of contributions.** We summarize our results below.

- We revisit the notion of edge centrality and provide a new interpretation as the gradient of the largest eigenvalue of $W^\top W$. The connection implies a generalization of edge centrality measures as the gradient of the sum of the largest $r$ eigenvalues of $W^\top W$.
- We develop an algorithm to reduce the number of infections in an epidemic spreading process by minimizing the sum of the largest $r$ eigenvalues of $W^\top W$, subject to an edge-weight reduction budget. Our algorithm runs a linear-time greedy selection step in an inner loop and provably converges to the global minimum.
- Our algorithm reduces the number of infections and the largest singular value of $W$ more than various baselines on an extensive collection of weighted networks, including mobility networks.

## 2 PRELIMINARIES

We review the SEIR model and its metapopulation extension. Then, we formulate a constrained optimization problem as our approach to mitigate an epidemic spreading process on weighted networks.

## 2.1 Epidemic Models

A widely used model of epidemic spread is the SEIR compartmental model (Durrett, 2007, Easley et al., 2010). An SEIR model uses four compartments to capture a spreading process: Susceptible (S), Exposed (E), Infected (I), and Recovered (R). Every node must belong to one of the four states during the process. At every time $t$,

- $S^{(t)}$ denotes the set of susceptible nodes at time $t$. A node may get exposed if its incoming neighbors are infectious. The probability depends on the edge weights and the virus transmission rate.
- $E^{(t)}$ denotes the nodes who have been exposed to the virus but who are not infectious at time $t$. In expectation, a node remains exposed for $\delta_E$ periods.
- $I^{(t)}$ denotes the nodes who are infectious at time $t$. Each node remains infectious for $\delta_I$ periods in expectation.
- $R^{(t)}$ denotes the nodes who have recovered at time $t$.

A metapopulation SEIR model is introduced in the mobility-based modeling approach of Chang et al. (2021a). The metapopulation SEIR model is launched on mobility networks. Mobility networks are bipartite graphs to model the traffic between population groups and locations. One part of the graph includes census block groups (CBGs), which involve a population of individuals in each group. The other part of the graph includes points of interest (POIs), which map to locations. Since there is a population of individuals in each CBG, one SEIR model is instantiated for each CBG. One of their key findings is that fitting the above metapopulation dynamics on mobility traffic data results in a surprisingly accurate prediction of the reported number of infected cases.

## 2.2 Optimization of Spectral Properties

Given a spreading process on a weighted graph, we are interested in designing algorithms to reduce the number of infected nodes. We focus on edge-weight reduction algorithms. As a motivating example, reducing the weight of an edge between a group and a location in mobility networks corresponds to restricting mobility.

Eigenvalue minimization is an approach to mitigate the spread of an epidemic process by removing edges (Tong et al., 2012). Let $G = (\mathcal{V}, \mathcal{E})$ be a weighted and possibly directed graph. Let $\mathcal{V}$ be the set of vertices and $\mathcal{E}$ be the set of edges. We use $W$ to denote a non-negative weight matrix over the edges. Let $W_{i,j}$ be the $(i, j)$-th entry of $W$. We extend the eigenvalue minimization approach to weighted networks as follows.

**Problem statement.** Given an edge weight reduction budget $B$, how can we modify the weight matrix $W$, so that the sum of the top-$r$ eigenvalues of $W^\top W$ is minimized? We will state a mathematical optimization formulation of this problem. Let $\lambda_i(W)$ be the $i$-th largest singular value of $W$, for $i$ from 1 to $r$. Notice that the square of $\lambda_i(W)$, denoted as $\lambda_i^2(M)$, is equal to the $i$-th largest eigenvalue of $M^\top M$. Thus, there is a one-to-one mapping between the singular values of $M$ and the eigenvalues of $M^\top M$, and they can be deduced from each other. Given a rank parameter $r$, we can formally state our problem as follows:

$$f^{\text{OPT}} \leftarrow \min_{M \in \mathbb{R}^{n \times m}} \quad f(M) := \sum_{k=1}^{r} \lambda_k^2(M) \tag{1}$$

$$\text{s.t.} \quad \sum_{(i,j) \in \mathcal{E}} \left( W_{i,j} - M_{i,j} \right) \leq B$$

$$0 \leq M_{i,j} \leq W_{i,j}, \text{ for any } (i, j) \in \mathcal{E}$$

$$M_{i,j} = 0, \qquad \text{for any } (i, j) \notin \mathcal{E}.$$

The input-output behavior of problem (1) is specified below:

- **Input:** A weighted graph $G = (\mathcal{V}, \mathcal{E})$ with a weight matrix $W$; An edge-weight reduction budget $B > 0$.
- **Output:** A modified $n$ by $m$ weight matrix $M$ that creates the largest decrease in the largest eigenvalue(s) of $W^\top W$, subject to:
  - **Feasibility constraint**: $0 \leq M_{i,j} \leq W_{i,j}$, for all $(i, j) \in \mathcal{E}$.
  - **Budget constraint**: $\sum_{(i,j) \in \mathcal{E}} \left( W_{i,j} - M_{i,j} \right) \leq B$.

Notice that our objective in equation (1) differs from Tong et al. (2012) in that we include the top-$r$ eigenvalues. When $r = 1$, the optimization objective reduces to the eigen-optimization problem

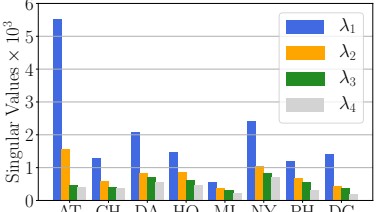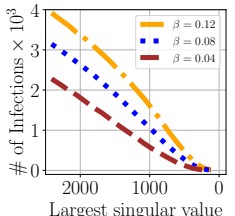

**Figure 2: (L) Largest four singular values of eight mobility networks. (R) Illustration of the connection between decreased largest singular values and decreased number of infections for various transmission rates ($\beta$) simulated on a mobility network.**

studied in Tong et al. (2012). Later in Section 4.3, we give an ablation study to justify our design choice.

**Illustration.** The largest singular value $\lambda_1$ is known to determine the epidemic threshold of graphs (Chakrabarti et al., 2008, Prakash et al., 2012). In Figure 2, we illustrate the connection between the largest singular value $\lambda_1$ and a SEIR diffusion process's spreading rate. As we scale down the graph's weight matrix, the number of infected nodes reduces. See Section 4.1 for the network dataset's description.

## 3 METHOD

We present a new algorithm to optimize problem (1) efficiently. To motivate our approach, we start by observing that the gradient of $f(M)$ is equivalent to the "edge centrality scores". Then, we develop an iterative algorithm with an inner loop that reduces edges with the highest edge centrality. We prove that our algorithm converges to the global optimum $f^{\text{OPT}}$, with a nearly-linear runtime in $|\mathcal{E}|$ per-iteration.

## 3.1 Edge Centrality

To motivate our approach, we begin by reviewing the algorithm of Tong et al. (2012), which considers controlling network diffusion by adding or removing edges. A central notion behind their approach is *edge centrality*, defined as the product of the *eigenvector scores* from both endpoints of an edge. More precisely, let $X$ be any matrix. Let $\vec{u}_1$ and $\vec{v}_1$ be the left and right singular vector of $X$, corresponding to $\lambda_1(X)$. Then, for any edge $(i, j) \in \mathcal{E}$, the edge centrality score of this edge is given by $\vec{u}_1(i) \cdot \vec{v}_1(j)$, where $\vec{u}_1(i)$ denotes the $i$-th coordinate of $\vec{u}_1$ and $\vec{v}_1(j)$ denotes the $j$-th coordinate of $\vec{v}_1$.

The edge-weight reduction can be viewed as a "continuous relaxation" of edge removal since the weight of an edge can be reduced by a fraction. Interestingly, we show that the edge centrality scores are equal to the gradient of the largest eigenvalue of $W^\top W$ (up to scaling), $\lambda_1^2(W)$, over decreasing the edge weights. A more general statement holds for a generalized notion of edge centrality scores, including the largest-$k$ singular values of $W$.

LEMMA 1 (INTERPRETING EDGE CENTRALITY AS THE GRADIENT OF LARGEST EIGENVALUES). *Assume that the singular values of $X$ are*

*all distinct. Then, the partial derivative of $\lambda_1^2(X)$ w.r.t. $X_{i,j}$ satisfies*

$$\frac{\partial \lambda_1^2(X)}{\partial X_{i,j}} = 2\lambda_1(X) \cdot \vec{u}_1(i) \cdot \vec{v}_1(j). \tag{2}$$

*More generally, for any $r \geq 1$, we have*

$$\frac{\partial \left( \sum_{k=1}^{r} \lambda_k^2(X) \right)}{\partial X_{i,j}} = 2 \sum_{k=1}^{r} \lambda_k(X) \cdot \vec{u}_k(i) \cdot \vec{v}_k(j). \tag{3}$$

PROOF. Consider a singular value $\lambda_k$ of $X$, for any $k$. Let $\vec{u}_k$ and $\vec{v}_k$ be the left and right singular vectors of $X$ corresponding to $\lambda_k$, respectively. By the chain rule, it suffices to show that $\frac{\partial \lambda_k(X)}{\partial X_{i,j}} = \vec{u}_k(i) \cdot \vec{v}_k(j)$. First, we have $\vec{u}_k^\top X = \lambda_k \vec{v}_k^\top$. We differentiate over $X$ on both sides of the above equation:

$$d(\vec{u}_k^\top)X + \vec{u}_k^\top \, d(X) = d(\lambda_k)\vec{v}_k^\top + \lambda_k \, d(\vec{v}_k^\top). \tag{4}$$

Since $\vec{v}_k$ is a unit length vector,

$$d(\|\vec{v}_k\|^2) = 2\langle \vec{v}_k, d(\vec{v}_k) \rangle = 2 \, d(\vec{v}_k^\top)\vec{v}_k = 0. \tag{5}$$

Thus, by multiplying both sides of equation (4) with $\vec{v}_k$, we get

$$d(\vec{u}_k^\top)X\vec{v}_k + \vec{u}_k^\top \, d(X)\vec{v}_k = d(\lambda_k)\vec{v}_k^\top \vec{v}_k + \lambda_k \, d(\vec{v}_k^\top)\vec{v}_k, \tag{6}$$

which is equal to $d(\lambda_k)$ since equation (5) holds and $v_k$ is a unit length vector. Looking at equation (6), we observe

$$d(\vec{u}_k^\top)X\vec{v}_k = d(\vec{u}_k^\top)\lambda_k \vec{u}_k = \lambda_k \, d(\vec{u}_k^\top)\vec{u}_k = 0, \tag{7}$$

where the last step follows similarly to equation (5), since $\vec{u}_k$ is also a unit length vector. In summary, we have shown $\vec{u}_k^\top \, d(X)\vec{v}_k = d(\lambda_k)$. This implies that the derivative of $\lambda_k$ over $X_{i,j}$ is equal to $\vec{u}_k(i) \cdot \vec{v}_k(j)$. Since this holds for any $k$, we thus conclude that equations (2) and (3) are both true. □

Given a weight matrix $W$ of a network, we compute the edge centrality scores via the best rank-$r$ approximation of $W$ as $\tilde{W}_r = U_r D_r V_r^\top$. More precisely, $D_r$ is an $r$ by $r$ square matrix, containing the largest $r$ singular values of $W$; $U_r$ is an $n$ by $r$ matrix, containing the left singular vectors corresponding to $D_r$; $V_r^\top$ is an $r$ by $m$ matrix, containing the right singular vectors corresponding to $D_r$. For every edge $(i,j) \in \mathcal{E}$, let $\tilde{W}_r(i,j)$ be the edge centrality score of this edge.

## 3.2 Iterative Edge Centrality Minimization

We now develop the *Frank-Wolfe-EdgeCentrality* minimization algorithm, or Frank-Wolfe-EC, specified in Algorithm 1. The high-level idea is iteratively applying a greedy selection of edges with the highest edge centrality scores while recomputing the scores. The input-output behavior of Frank-Wolfe-EC is as follows:

- **Input:** The primary inputs are graph $\mathcal{G}$, budgeted reduction amount $B$, and rank-$r$ in the objective $f(M)$ (cf. equation (1)).
- **Output:** A weight matrix $M$ with reduced edge weights.

The algorithm also requires two parameters, the number of iterations $T$ and a range of learning rates $H$. At every iteration $t$ from 1 to $T$, let $M_t \in \mathbb{R}^{n \times m}$ be the currently modified weight matrix. Let $\nabla f(M_t)$ be the gradient of $f(M_t)$. We will apply the Frank-Wolfe algorithm (Frank and Wolfe, 1956, Nocedal and Wright, 2006), which is an iterative approach for constrained minimization problems.

The Frank-Wolfe algorithm computes a descent direction $G_t$ for $M_t$, that minimizes the following matrix inner product

$$\langle \nabla f(M_t), G_t \rangle = \text{Tr}\left[ \nabla f(M_t)^\top G_t \right],$$

subject to the same set of constraints as problem (1):

$$G_t^\star \leftarrow \arg\min_X \quad \langle X, \nabla f(M_t) \rangle \tag{8}$$

$$\text{s.t.} \quad \sum_{(i,j) \in \mathcal{E}} \left( W_{i,j} - X_{i,j} \right) \leq B$$

$$0 \leq X_{i,j} \leq W_{i,j}, \text{ for any } (i,j) \in \mathcal{E}$$

$$X_{i,j} = 0, \qquad \text{for any } (i,j) \notin \mathcal{E}.$$

Thus, problem (8) minimizes the matrix inner product between $G_t$ and $\nabla f(M_t)$, subject to the same set of constraints as problem (1).

**Greedy selection.** The core of our approach is to show that the optimal descent direction, $G_t^\star$, is given by a greedy selection of edges based on their edge centrality scores. This procedure, Top-K-EdgeCentrality, or Top-K-EC, is specified as part of Algorithm 1. Let $\tilde{W}_r^{(t)}$ be the best rank-$r$ approximation of $M_t$ for every $t$. Let $(i_1, j_1), (i_2, j_2), \ldots, (i_m, j_m)$ be the edges in descending order of their edge centrality scores $\tilde{W}_r^{(t)}$, where $m = |\mathcal{E}|$ is the number of edges. Consider the first $k$ edges whose total weight exceeds the reduction budget $B$:

$$\sum_{l=1}^{k-1} W_{i_l,j_l} < B \text{ and } \sum_{l=1}^{k} W_{i_l,j_l} \geq B. \tag{9}$$

Then, the weight of the first $k-1$ edges is reduced to zero. The weight of the last edge decreases by $\sum_{l=1}^{k} W_{i_l,j_l} - B$. The following result proves that the above greedy selection yields an optimal solution of the inner optimization problem (8).

THEOREM 2 (OPTIMAL DESCENT DIRECTION IS GIVEN BY GREEDY SELECTION). *The optimal solution of Problem 8, $G_t^\star$, is given by the output of Top-K-EdgeCentrality($W, B; M_t$) (cf. Algorithm 1).*

PROOF. By Lemma 1, for every edge $(i,j) \in \mathcal{E}$, the gradient of $f(M_t)$ over this edge is given by the *edge centrality* scores. Since $X_{i,j} = 0$ for any $(i,j) \notin \mathcal{E}$, the optimization objective is:

$$\langle X, \nabla f(M) \rangle = \sum_{(i,j) \in \mathcal{E}} 2X_{i,j} \left( \sum_{k=1}^{r} \lambda_k \cdot \vec{u}_k(i) \cdot \vec{v}_k(j) \right). \tag{10}$$

Above, each variable $X_{i,j}$ is multiplied precisely by the edge centrality of the edge $(i,j)$ (cf. line (16)).

Consider minimizing the equivalent objective (10) with the constrains of Problem (8). The minimizer, $G_t^\star$, is achieved by reducing the weight of the edges with the highest edge centrality to zero until the budget $B$ gets exhausted. This is precisely the procedure of Top-K-EC from lines (16)-(19). Thus, we have proved this result. □

After finding the descent direction $G_t^\star$, the next step of the Frank-Wolfe algorithm is setting a learning rate by minimizing $f((1 - \eta_t)M_t + \eta_t G_t)$, for $\eta_t$ in a range $H$ between 0 and 1. Then, we update the weight matrix accordingly.

To conclude, each iteration of Frank-Wolfe-EC computes a truncated rank-$r$ SVD of a sparse matrix with at most $m$ nonzeros and sorts an array of size $m$. The former requires a runtime complexity

**Algorithm 1** Iterative Edge Centrality Minimization

**Input:** A graph $\mathcal{G} = (\mathcal{V}, \mathcal{E})$ with weight matrix $W$; Budget $B$.
**Output:** A weight matrix $M$ modified from $W$.
**Parameters:** Rank $r$; Number of iterations $T$; Range of learning rate $H$.

1: **procedure** FRANK-WOLFE-EDGECENTRALITY($W, B; T, H$)
2:      Let $M_0 = W$
3:      **for** $t = 0, 1, \ldots, T - 1$ **do**
4:          $G_t^\star$ = TOP-K-EDGECENTRALITY($W, B; M_t$)
5:          Set $\eta_t$ by minimizing $f\big((1 - \eta_t)M_t + \eta_t G_t^\star\big)$ for $\eta_t \in H$
6:          $M_{t+1} = (1 - \eta_t) \cdot M_t + \eta_t \cdot G_t^\star$
7:      **end for**
8:      **if** there is unused budget in $M_T$ **then**
9:          $B' = B - \text{sum}(W - M_T)$
10:         $M^\star$ = TOP-K-EDGECENTRALITY($M_T, B'; M_T$)
11:      **end if**
12:      **return** $M^\star$
13: **end procedure**
14:
15: **procedure** TOP-K-EDGECENTRALITY($W, B; M$)
16:      Let $\tilde{M}_r$ be the best rank-$r$ approximation of $M$
17:      Sort the edges in $\mathcal{E}$ by their edge centrality scores from $\tilde{M}_r$
18:      Let $k$ be the number of edges by Equation 9 with weight matrix $W$
19:      Reduce the first $k - 1$ edges' weight to zero and the last edge's weight by the remaining budget
20:      **return** the updated $W$ matrix
21: **end procedure**

---

of $O(mr \log(m))$ (e.g., Theorem 1 of Musco and Musco (2015)). The latter can be achieved with runtime $O(m \log(m))$. By comparison, the runtime complexity for solving a general linear program (i.e., problem (8)) is at least quadratic in the dimension of $W$ (Cohen et al., 2021).

**Running time guarantee.** Next, we examine the number of iterations that our algorithm needs to converge to $f^{\text{OPT}}$. A well-established result (e.g., Jaggi (2013)) is that for convex minimization problems, the Frank-Wolfe algorithm will converge to the global minimum under mild conditions. In the following, we will show that the objective $f(M)$ is convex. Based on that, we show that our Frank-Wolfe-EC algorithm will converge to the global minimum of problem (1), at a rate of $O(t^{-1})$ after $t$ iterations.

THEOREM 3 (CONVERGENCE RATE OF FRANK-WOLFE-EC). *Let $\kappa$ be the minimum of $\sigma_r(M_t) - \sigma_{r+1}(M_t)$ and $\mu$ be the maximum of $\sigma_1(M_t)$, from $t = 0, 1, \ldots, T - 1$. Assume that $\kappa$ is strictly positive. Let $\alpha_1 = 8 \sum_{(i,j) \in \mathcal{E}} W_{i,j}^2$ and $\alpha_2 = 4r + \frac{5\mu \cdot \sqrt{r}}{\kappa} + C$, for some absolute fixed constant $C > 0$. Then, we have the following approximation guarantee for $M_T$,*

$$f(M_T) - f^{\text{OPT}} \leq \frac{\alpha_1 \alpha_2}{T}. \tag{11}$$

Theorem 3 implies that our algorithm will eventually converge to minimize problem (1), under mild conditions on the spectral gap of the iterates. See Section B for the proof of Theorem 3. The constants $\alpha_1 \alpha_2$ can be large (as are previous guarantees of the Frank-Wolfe

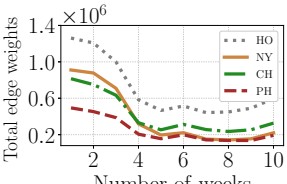 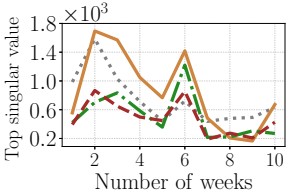

**Figure 3: Illustration of the mobility patterns over ten networks. (L) Total edge weights; (R) Largest singular value.**

algorithm). Though later in Section 4.3, we find that the number of iterations $T$ required for Frank-Wolfe-EC to converge is less than 30 on all network datasets.

**Extension.** Our study has focused on mitigating the spread in a static network. Another consideration is that network typologies evolve over time. Therefore, an important question is how to tackle such temporal evolution. We plot the total edge weights and the largest singular value of ten mobility networks over ten weeks to understand the evolving network patterns. See Section 4.1 for the data set description. Figure 3 shows the result: Interestingly, these four metropolitan statistical areas exhibit similar mobility patterns.

We extend the Frank-Wolfe-EC algorithm to temporal networks in two steps. Let $W_1, W_2, \ldots, W_T$ be the weight matrix of a sequence of networks. First, we allocate a certain budget to every network in the sequence. We allocate the budget proportional to each network's largest singular value. The budget for the $i$-th network is set as

$$\frac{\lambda_1(W_i)}{\sum_{j=1}^{T} \lambda_1(W_j)} \cdot B. \tag{12}$$

Second, we apply Frank-Wolfe-EC to every network with the allocated budget. Thus, such an allocation will prioritize reducing the edge weights of networks with higher singular values.[1]

## 4 EXPERIMENTS

We evaluate our proposed approaches on a range of mobility networks and weighted graphs. Our experiments seek to address the following three questions. (1) Does our proposed algorithm perform well compared to methods from prior works? (2) How well does the algorithm reduce the largest singular value of the network weighted matrix? (3) How does the temporal allocation extension of our algorithm perform in practice? We present positive results to answer these three questions, validating the practical benefit of our algorithm. The code for reproducing the experiments is available in an anonymous link. [2]

### 4.1 Experimental Setup

*4.1.1 Datasets.* We follow the procedure described within Chang et al. (2021a) to construct the mobility networks. We briefly summarize the procedure and defer a comprehensive discussion to their

---

[1]We remark that implementing this scheme requires knowing the largest singular value of every network in the sequence. When such information is not available, one needs first to estimate this information (e.g., by predicting such information based on previous network patterns (Qin et al., 2018, Wang and Tang, 2019)). This is left for future work.

[2]https://github.com/anonymous-researchcode/Iterative-Edge-Centrality-Min.

**Table 1: Basic statistics for eight mobility networks constructed from SafeGraph's monthly patterns data (Chang et al., 2021a).**

|  | AT | CH | DA | HO | MI | NY | PH | DC |
|---|---|---|---|---|---|---|---|---|
| Number of census block groups | 2,799 | 5,784 | 4,069 | 4,029 | 2,279 | 10,170 | 3,547 | 2,564 |
| Number of points of interest | 8,433 | 26,606 | 15,000 | 34,866 | 15,559 | 24,046 | 15,102 | 8,026 |
| Number of nonzero weighted edges | 154,729 | 439,262 | 283,928 | 671,217 | 276,109 | 463,719 | 260,279 | 107,733 |
| Average edge weight | 5.258 | 4.659 | 4.921 | 4.951 | 4.833 | 4.749 | 4.864 | 4.848 |
| Average population per group | 2400.402 | 1593.152 | 2069.406 | 2395.407 | 2219.864 | 1578.068 | 1568.618 | 2060.778 |

**Table 2: Basic statistics for three weighted graphs.**

|  | Airport | Advogato | Bitcoin |
|---|---|---|---|
| Number of Nodes | 7,977 | 6,541 | 3,783 |
| Number of Edges | 30,501 | 51,127 | 24,186 |
| Average edge weight | 1.45 | 0.83 | 1.46 |

paper. The construction requires the following data sources: (i) Mobility patterns from the Monthly Pattern and Weekly Pattern datasets (SafeGraph, 2020a,b), (ii) The geometry dataset (SafeGraph, 2021), and (iii) The Open Census Dataset (SafeGraph, 2018). Additionally, we will use the reported cases of COVID-19 infections from The New York Times to calibrate the SEIR model. We generate the mobility networks based on monthly patterns from March 2, 2020, to May 10, 2020. We report the statistics of the mobility networks from each metropolitan statistical area (MSA) in Table 1. Overall, the mobility patterns cover 25,341 CBGs with over 65 million people and 147,638 POIs. The temporal mobility networks are constructed based on weekly mobility patterns during the same period mentioned above, including ten networks for every MSA.

Besides mobility networks, we consider three other weighted networks: (i) An Airport traffic network of flights among all commercial airports in the world (Opsahl, 2011); (ii) A network of trust relationships among users on Advogato; (Massa et al., 2009); (iii) A network of trust relationships among users on the Bitcoin Alpha platform. (Kumar et al., 2016). Edge weights in the last two networks denote different levels of declared trust among users. The statistics of these three networks are listed in Table 2.

*Data availability.* The mobility data is freely available to researchers, non-profit organizations, and governments through the SafeGraph COVID-19 Data Consortium.[3] The New York Times COVID-19-data is publicly available online.[4] Links to the other weighted networks are included in the references.

*4.1.2 Baseline methods.* The experiments for spreading on a static network involve the following baseline methods.

- K-EdgeDeletion: Delete a set of edges with the highest edge centrality scores according to the best rank-1 approximation of $W$, subject to the reduction budget (Tong et al., 2012).
- Edge weighted reduction: Reduce the weight of every edge by a ratio that is proportional to its weight.
- Uniform reduction: Uniformly reduce the weight of every edge by the same fraction, subject to the budget constraint.

---

[3]https://www.safegraph.com/covid-19-data-consortium
[4]https://github.com/nytimes/covid-19-data

- Max occupancy capping: Reduce the cumulative weights at each POI proportional to its max occupancy.
- Capping by POI category: Cap the maximum occupancy of a particular category of POIs.

The last three baselines are adapted from Chang et al. (2021a).

For the experiments on a sequence of temporal networks, we will only use Algorithm 1 to modify the network weight matrix while varying the budget allocation scheme. We consider the following list of allocation schemes along with the scheme from equation 12.

- First week only: Assign all the edge-weight reduction budget to the first week of the sequence.
- Uniform allocation: Distribute the budget uniformly among every network in the sequence.
- Exponential allocation: Distribute the budget proportional to $\exp(-t)$, decaying exponentially over time.

*4.1.3 Implementation.* For the experiments concerning mobility networks, we follow the procedures of Chang et al. (2021a) to simulate a metapopulation SEIR model in each network. We simulate 100 epochs to be consistent with the simulation of Chang et al. (2021a). The results are consistent throughout the simulation. We compare the FRANK-WOLFE-EC algorithm with baseline methods using an edge-weight reduction budget as 5% of the total edge weights. Results of using other budget amounts are consistent.

We simulate an SEIR model on each graph for the other weighted networks. To avoid infecting all the graph nodes, we simulate for 50 epochs. We use a slightly higher edge-weight reduction budget as 20% of the total edge weights because the average edge weight in these three graphs is smaller than the mobility networks.

For the temporal mobility networks experiments, we simulate the metapopulation SEIR model on a sequence of ten networks for 70 epochs or seven epochs for every network. We set the edge-weight reduction budget as 5% of the total edge weights of the sequence and allocate the budget to each network by allocation strategies described above.

We calibrate the parameters of the SEIR model following the method presented in Chang et al. (2021a). Specifically, we calibrate the following parameters: (i) the transmission constant in POIs, $\psi$; (ii) the base transmission rate, $\beta_{\text{base}}$; and (iii) the ratio of initial exposed people, $p_0$. We use grid search to find the parameters with the smallest root mean square error compared to the reported number of infected cases. We calibrate an SEIR model for every MSA independently. For the weighted graphs, we use a transmission rate $\beta_{\text{Base}} = 0.05$ and a initial exposed ratio $p_0 = 0.01$.

In Algorithm 1, we search the rank parameter $r$ in $[1, 50]$ and the number of iterations in $[5, 30]$. For each result reported in Section

**Table 3: Comparison of Frank-Wolfe-EC to baseline methods on mobility networks. We modify the edge weights using the strategy in each row. Top: The total number of infected populations ($\times 10^3$). Bottom: Comparison of the largest singular value. Results are averaged over 50 runs.**

| Infected populations | Atlanta | Chicago | Dallas | Houston | Miami | New York | Philadelphia | Washington DC |
|---|---|---|---|---|---|---|---|---|
| No Intevention | 48.45±3.17 | 1858.81±46.53 | 91.91±21.29 | 366.55±26.74 | 752.53±26.94 | 3146.57±21.43 | 492.52±20.75 | 41.10±2.20 |
| Uniform Reduction | 46.80±2.37 | 1762.01±64.38 | 84.76±11.14 | 312.11±26.31 | 671.32±23.72 | 2996.90±40.06 | 463.42±12.80 | 41.11±1.64 |
| Weighted Reduction | 43.22±2.77 | 782.53±86.92 | 66.18±3.29 | 194.636±18.58 | 43.42±12.64 | 1336.65±60.05 | 342.19±10.46 | 40.76±1.36 |
| Max Occ. Capping | 44.38±2.70 | 1741.16±65.30 | 82.34±8.67 | 315.34±33.37 | 675.56±26.52 | 2990.03±45.23 | 455.15±15.90 | 41.53±1.70 |
| POI Category | 46.17±3.20 | 1728.66±62.58 | 77.37±8.47 | 283.87±31.66 | 687.65±25.52 | 2950.27±38.45 | 458.36±17.62 | 41.04±1.44 |
| K-EdgeDeletion | 44.92±2.96 | 346.86±40.64 | 64.11±2.88 | 186.56±18.99 | 78.29±8.96 | 352.92±27.70 | 185.22±10.64 | 39.85±0.91 |
| Top-k-EC | 45.82±3.52 | 355.19±46.55 | 64.06±2.41 | 187.24±21.25 | 78.91±7.94 | 362.91±36.31 | 178.62±11.28 | 39.92±1.29 |
| Frank-Wolfe-EC | **40.47±1.78** | **166.23±16.17** | **62.07±2.47** | **86.40±10.98** | **8.53±2.58** | **301.46±88.41** | **129.23±13.4 8** | **39.27±1.06** |
| Largest singular value | Atlanta | Chicago | Dallas | Houston | Miami | New York | Philadelphia | Washington DC |
| No Intevention | 5526.633 | 1296.219 | 2093.614 | 1467.722 | 555.780 | 2413.421 | 1203.220 | 1406.572 |
| Uniform Reduction | 5250.302 | 1231.409 | 1988.933 | 1394.337 | 527.991 | 2292.751 | 1143.059 | 1336.244 |
| Weighted Reduction | 1254.204 | 302.670 | 564.700 | 420.529 | 213.059 | 481.870 | 374.599 | 365.950 |
| Max Occ. Capping | 5250.302 | 1231.409 | 1988.933 | 1394.337 | 527.991 | 2292.751 | 1143.059 | 1336.244 |
| POI Category | 5526.390 | 1295.837 | 2073.784 | 1467.647 | 555.619 | 2270.005 | 1202.895 | 1375.490 |
| K-EdgeDeletion | 1565.267 | 257.987 | 417.208 | 447.988 | 216.761 | 355.909 | 282.725 | 227.280 |
| Top-k-EC | 1565.264 | 257.987 | 417.192 | 447.987 | 216.761 | 355.902 | 282.720 | 226.879 |
| Frank-Wolfe-EC | **1191.216** | **125.901** | **308.401** | **235.638** | **169.539** | **197.262** | **190.066** | **188.135** |

**Figure 4: Illustration of the results for the three weighted graphs. (4a): Number of infected nodes from simulating an SEIR model on the modified weight matrix averaged over 50 runs. (4b): The largest singular value of the modified weight matrix.**

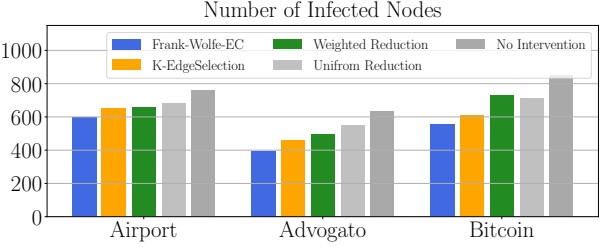

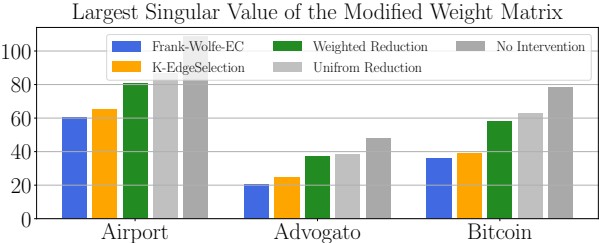

**(a) Number of infected nodes from SEIR model simulations.**   **(b) Largest singular value of modified network weight matrix.**

4, we search the two hyper-parameters 50 times. For the range of learning rate $H$, we use 30 values from the range of $[10^{-3}, 10^{-1}]$ as $H$. In each iteration of the algorithm, we conduct a grid search over the learning rate range and choose the learning rate $\eta_t$ that leads to the smallest object value $f\big((1 - \eta_t)M_t + \eta_t G_t^\star\big)$. All the experiments are conducted on an AMD 24-Core CPU machine.

## 4.2 Experimental Results

**Does Algorithm 1 reduce the number of infected nodes effectively?** We begin by comparing the number of infected nodes between our algorithm and the baseline methods.

*Results for weighted graphs:* Figure 4a compares our algorithm to baseline intervention strategies on three weighted graphs. Overall, we see that Frank-Wolfe-EC reduces the number of infected nodes by **10.46**% more than baseline methods on average.

*Results for mobility networks:* Table 3 compares the total number of infected populations using different intervention strategies on eight networks. We note that Frank-Wolfe-EC—our iterative optimization method—outperforms other baselines by **30.17**% on average and up to **80.36**%. Additionally, we observe that the trend is consistent with Table 3 during the entire spreading process.

*Similar results with different budgets:* We have also observed similar results by varying the budget for mobility reduction. We vary the budget from 1% to 20% using the New York mobility network. We find that our algorithm outperforms the baseline methods consistently using different budget levels, similarly for the largest singular value. Interestingly, when the level of budget is small (e.g., 1%), Frank-Wolfe-EC reduces the largest singular value more significantly than baseline methods.

**How much does the largest singular value decrease?** Next, we report the drop in the largest singular value. Figure 4b illustrates the largest singular value of the modified weight matrix of the three weighted graphs. Frank-Wolfe-EC reduces the largest singular value more than baselines by **11.42**% on average. Additionally, Table 3 reports the largest singular value of networks modified by each edge-weight reduction strategy on mobility networks. Frank-Wolfe-EC is **30.75**% more effective than the best baseline on average.

**Is the extension to temporal weighted networks effective?** Finally, we evaluate our algorithm and a budget allocation strategy over a sequence of weighted temporal networks. We find that allocating the budget to every network proportional to their largest

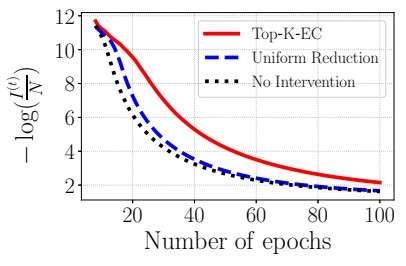

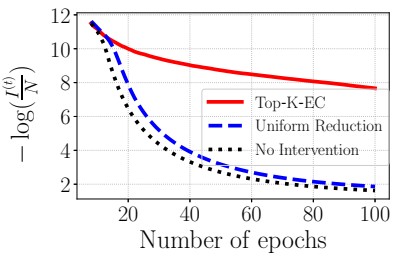

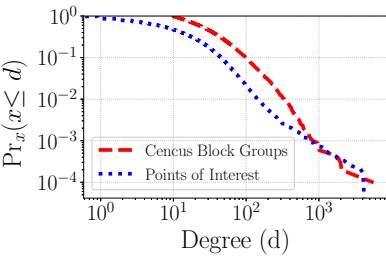

**(a) Budget $B$ = 1% of total edge weights.** **(b) Budget $B$ = 20% of total edge weights.** **(c) CCDF of CBGs and POIs.**

**Figure 5: Comparison of greedy selection and uniform edge-weight reduction on a mobility network. Top-K-EC is more effective in reducing the infected proportion throughout the SEIR model simulation. Moreover, the CBGs and the POIs in this network have a heavy-tailed degree distribution, which supports the intuition behind selecting edges by their centrality scores.**

singular value outperforms all the other allocations. In particular, the number of infected populations is smaller by **39.82%** averaged over both Chicago and New York mobility networks.

*Model validation:* We compare the predicted cases of our simulated SEIR model with the reported cases from New York Times COVID-19 data. The root mean squared error of all the epochs is 295.17 averaged over eight mobility networks. The error is within 3% compared to the overall infected population which is at the scale of $10^4$. These results reaffirm the finding of Chang et al. (2021a).

### 4.3    Ablation Study

**Benefit of choosing rank $r$.** Recall that our algorithm requires specifying rank $r$–the number of top singular values–in Equation 1. We hypothesize that varying the rank $r$ would lead to different intervention results. We ablate the performance of our algorithm by using different $r$ in a range of $[1, 50]$. The results show that the performance of the best choice $r$ outperforms using $r = 1$ by 40.27% averaged over all networks. This result justifies our formulation of the network intervention problem as an optimization for the sum of largest-$r$ singular values instead of only the largest single value.

**Choosing edges via edge centrality.** We validate that removing edges via top edge centrality scores effectively reduces infections. Figure 5 compares Top-K-EC (cf. Algorithm 1) to uniformly reducing every edge's weight by the same ratio. With Top-K-EC, the largest singular value dropped by 57.7% in (5a) and 96.8% in (5b). With uniform reduction, the drop goes down to 1% and 20%, respectively.

**Benefit of the iterative approach.** The greedy selection algorithm Top-k-EC can be viewed as a special case of Frank-Wolfe-EC with $T = 1$. Notice that the iterative approach is necessary to get the observed empirical performance. In Table 3, Frank-Wolfe-EC outperforms Top-k-EC by **31.41%** on average, and the largest singular value is reduced by **33.09%** more in Table 3.

## 5    DISCUSSION AND RELATED WORK

There is an extensive body of work studying diffusion control on networks. Besides epidemic spreading, network diffusion is also widely studied in social and information networks (Goel et al., 2015, 2016, Matsubara et al., 2012). We summarize the most relevant research to ours while referring the reader to Pastor-Satorras et al.

(2015)'s survey for references. A key result in the epidemics literature is that the largest eigenvalue of the adjacency matrix (a.k.a. the spectral radius, denoted as $\lambda_1$) characterizes the epidemic threshold for more than 25 propagation models (Prakash et al., 2012).

An important implication of this result is that the epidemic dies out if $\lambda_1$ decreases, and this is the basis of many works on epidemic control (Chen et al., 2016, Le et al., 2015, Torres et al., 2021, Van Mieghem et al., 2011). Because eigen-optimization problems via edge additions or deletions are NP-hard (Khalil et al., 2014, Tong et al., 2012), approximation algorithms are used for diffusion control. A key approach is a greedy algorithm based on some notion of centrality information in the network (Parotsidis et al., 2016). There is a connection between this problem and submodular optimization, leading to provable approximation ratios for the greedy algorithm (Chen et al., 2018, Saha et al., 2015).

Besides, there is a line of work studying diffusion control under the name of the Firefighter problem in approximation algorithms (Anshelevich et al., 2009, Finbow and MacGillivray, 2009). Finally, there are studies on the design of vaccine distribution for pandemic control (Sambaturu et al., 2020, Zhang and Prakash, 2014). These works and their analysis do not lead to direct bounds for weighted networks, which is the focus of our setting.

The modeling of dynamical processes in social-technical systems requires both data-driven models and theoretical understanding of the dynamics (Mønsted et al., 2017, Vespignani, 2012). Besides SEIR compartmental models, there are other ways to model network spreading processes. The critical algorithmic insight of our work is to strategically restrict mobility using spectral properties of a network. While our study focuses on the SEIR model and applications to mobility-based modeling, it is conceivable that our algorithmic insights might apply to different epidemic models and different data-driven modeling of the pandemic. For example, an interesting research question is to examine our approach with different epidemic models such as SIS and SIR. Besides, another interesting question is to study node deletion as the intervention. In the context of mobility networks, reducing the weight of a node means reducing a fraction of the node's mobility. Lastly, it is conceivable that one can combine our approach with existing techniques to better deal with temporal dynamics (Prakash et al., 2010). These questions are left for future work.

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

## A  RUNTIME RESULT

We report the runtime of FRANK-WOLFE-EC. Across all eleven networks, the FRANK-WOLFE-EC converges within 30 iterations (17 iterations on average). At each iteration, the SVD step takes less than 3 seconds. The other steps in the inner loop take less than 2.7 seconds on all eleven networks.

Next, we report the per-iteration runtime of FRANK-WOLFE-EC on graphs with 1.5 million to 117 million edges. In addition to the 11 networks, we run our method on seven graphs, including com-Orkut (with 117M edges), com-LiveJournal (34M), wiki-topcats (28M), web-BerkStan (7.6M), web-Google (5.1M), web-Stanford (2.3M), and web-NotreDame (1.4M) from the SNAP datasets (Leskovec and Krevl, 2014). Figure 6 illustrates the per-iteration runtime on these graphs (including the eleven networks described above). Notice that the runtime scales are nearly-linear with the number of edges: The slope is less than 1 in Figure 6, which means the runtime is less than linear in the size of the graph. For example, our algorithm takes 4943 seconds per iteration on the largest graph with 117 million edges and 3 million nodes. These results show that our algorithm runs efficiently on large-scale graphs.

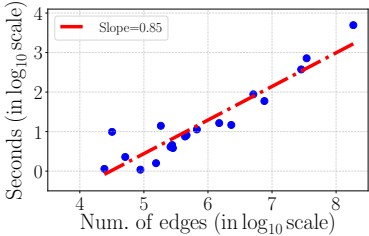

**Figure 6: Runtime of Frank-Wolfe-EC in log-log scale for one iteration. The number of edges ranges from $10^4$ to $10^8$ and the number of nodes ranges from $10^3$ to $10^6$.**

## B  PROOF OF THEOREM 3

We complete the convergence analysis of our algorithm. First, we show that the objective function $f(M)$ is convex in $M$. Second, we invoke the result of Jaggi (2013), specifically Lemma 7 and Theorem 1, which show that as long as the gradient $\nabla f(M)$ is Lipschitz-continuous and the constraint set has bounded diameter, the Frank-Wolfe algorithm will converge to the optimum at a rate of $O(\frac{1}{t})$ after $t$ iterations.

PROOF OF THEOREM 3. We first show that the sum of top singular values $g(M) = \sum_{k=1}^{r} \lambda_k(M)$ is convex. With the variational characterization of singular values, $g(M)$ is equal to

$$g(M) = \max_{U^\top U = V^\top V = \mathrm{Id}_r: \, U \in \mathbb{R}^{n \times r}, V \in \mathbb{R}^{m \times r}} \langle UV^\top, M \rangle. \quad (13)$$

Thus, for any $n$ by $m$ matrix $M_1, M_2$, and any $\alpha \in [0,1]$, let $\tilde{U}$ and $\tilde{V}$ be the maximizer of the above for $f(\alpha M_1 + (1-\alpha)M_2)$. Therefore,

$$g(\alpha M_1 + (1-\alpha)M_2) = \langle \tilde{U}\tilde{V}^\top, \alpha M_1 + (1-\alpha)M_2 \rangle$$
$$\leq \alpha \langle \tilde{U}\tilde{V}^\top, M_1 \rangle + (1-\alpha)\langle \tilde{U}\tilde{V}^\top, M_2 \rangle$$
$$\leq \alpha g(M_1) + (1-\alpha)g(M_2),$$

which implies that $g(M)$ is convex. Next, we show that $f(M)$ is convex. For any $\alpha \in [0,1]$,

$$f(\alpha M_1 + (1-\alpha)M_2) = g\left((\alpha M_1 + (1-\alpha)M_2)^T (\alpha M_1 + (1-\alpha)M_2)\right)$$
$$\leq \alpha^2 g(M_1^\top M_1) + (1-\alpha)^2 g(M_2^\top M_2) + 2\alpha(1-\alpha)g(M_1^\top M_2).$$

Let $\tilde{U}$ and $\tilde{V}$ be the maximizer of (13) for $M_1^\top M_2$. We have

$$2g(M_1^\top M_2) = 2\langle \tilde{U}\tilde{V}^\top, M_1^\top M_2 \rangle = 2\langle M_1\tilde{U}, M_2\tilde{V} \rangle$$
$$\leq \left\|M_1\tilde{U}\right\|_F^2 + \left\|M_2\tilde{V}\right\|_F^2 = \langle M_1^\top M_1, \tilde{U}\tilde{U}^\top \rangle + \langle M_2^\top M_2, \tilde{V}\tilde{V}^\top \rangle$$
$$\leq g(M_1^\top M_1) + g(M_2^\top M_2).$$

Therefore, $f(\alpha M_1 + (1-\alpha)M_2)$ is less than $\alpha \cdot g(M_1^\top M_1) = \alpha \cdot f(M_1)$ plus $(1-\alpha) \cdot g(M_2^\top M_2) = (1-\alpha) \cdot f(M_2)$.

Second, we verify that $\nabla f(M)$ is $\alpha_2$ Lipschitz continuous in the Frobenius norm. The proof is based on matrix perturbation bounds. Let $\tilde{M} = M + E$ be a perturbation of $M$. Let $M_r = U_r D_r V_r^\top$ be the top-$r$ SVD of $M$. Let $\mu_1$ be the largest singular value of $M$. Let $\tilde{M}_r = \tilde{U}_r \tilde{D}_r \tilde{V}_r^\top$ be the top-$r$ SVD of $\tilde{M}$. First, consider $\|E\|_2 \leq \kappa/2$. By matrix perturbation bounds on the truncated SVD of a matrix (e.g., Theorem 1 of Vu et al. (2021); the condition is satisfied since $\kappa$ is the spectral gap between the $r$-th and $(r+1)$-th largest singular values), we have

$$\|M_r - \tilde{M}_r\|_F^2 \leq 2\|E\|_F^2 + \frac{4\lambda_1^2}{\kappa^2}\|E\|_F^2 + C\|E\|_F^2.$$

When $\|E\|_2 \geq \kappa/2$, notice that

$$\|M_r - \tilde{M}_r\|_F^2 = \|U_r D_r V_r^\top - \tilde{U}_r \tilde{D}_r \tilde{V}_r^\top\|_F^2$$
$$\leq 2\|D_r\|_F^2 + 2\|\tilde{D}_r\|_F^2$$
$$\leq 2r\lambda_1^2 + 2r(\lambda_1 + \|E\|_2)^2,$$

which is at most $2r(3\lambda_1^2 + 2\|E\|_2^2)$. The step above uses the Weyl's Theorem that $\|D_r - \tilde{D}_r\|_2 \leq \|E\|_2$. Taken together, we conclude that $\nabla f(M)$ must be

$$\sqrt{\max\left(2 + \frac{4\lambda_1^2}{\kappa^2} + C, \frac{24r \cdot \lambda_1^2}{\kappa^2} + 4r\right)}$$

Lipschitz continuous. Lastly, the diameter of the constraint set is at most $\sqrt{\sum_{(i,j)\in\mathcal{E}} W_{i,j}^2}$, since for every $(i,j) \in \mathcal{E}$, the search space is bounded between 0 and $W_{i,j}$. Taken together, we have proved that: $f(M)$ is convex, $\nabla f(M)$ is $\alpha_2$ Lipschitz continuous, and the diameter of the constrained space of problem (1) is $\sqrt{\alpha_1/8}$. Using Lemma 7 and Theorem 1 of Jaggi (2013), the proof is complete.  □

## C  CONCLUSION AND FUTURE WORK

We studied the problem of controlling the diffusion of an epidemic on weighted networks via reducing edge weights. This problem is motivated by recent studies of mobility-based modeling for the COVID-19. We introduced a constrained optimization problem to reduce edge weights that minimize the network's largest singular values. We designed an iterative procedure for finding the global minimum of the above optimization problem. Our algorithm is guaranteed to converge to the global optimum. Additionally, we theoretically proved and empirically observed that each iteration only requires a nearly-linear runtime in the size of the network.

Our experiments demonstrated the superiority of our approaches. Our work highlights the existence of spectral properties in mobility networks and uses them to design practical intervention algorithms.

We mention two questions for future work. First, although we demonstrated that choosing the rank $r$ larger than one yields superior empirical performance, theoretically justifying the reduction of $f(M)$ to reducing epidemic threading is still lacking, and we leave it for future work. Second, while our algorithm achieved strong empirical performance on mobility networks, theoretically analyzing it within the setting of mobility-based modeling remains an interesting question. In particular, we are not aware of any result on the epidemic threshold of the metapopulation SEIR model. More broadly, we hope our work inspires further algorithmic and theoretical studies on epidemic spreads, which could in turn contribute to the ongoing discussion of pandemic prevention.