# OpenReview forum: "Optimal Intervention on Weighted Networks via Edge Centrality"
_ACM.org/SIGKDD/2022/Workshop/epiDAMIK — KDD 2022 Workshop epiDAMIK Oral_

### Official Review · Reviewer_BNij · 2022-06-25
**Paper 10 clearly motivates the topic and guides the reader through a comparison to previous methods.**

**Rating:** 3
**Confidence:** 3

**Review:**

This paper seeks to find intervention strategies for epidemics by reducing weights in a contact network. This may reflect an isolation strategy the best, which curtails an outbreak by reducing contacts between individuals.  This is done by decreasing edge weights in the contact graph. Previously, edge weight have been eliminated in an “all or nothing” fashion where certain weights are removed completely. This procedure decreases the weights as a relaxation of the previous problem.

__Quality__: Well written overall with excellent comparison to previous studies.

__Significance__: Improvements on previous studies are fairly large, so this method has promise.

__Clarity__: The introduction and conclusion are very well laid out, but some of technical sections are hard to follow.

__Originality__: This method is fairly original, but not completely novel as it is a natural extension of its previous studies where $r = 1$

__Strong points__
 - Well motivated – previous research is outlined in a helpful manner
 - Subtle details in definitions are clearly stated.
 - The structure of the experimental results section is helpful, especially for explaining the important of each part of the approach.
 - The consistency with methods of previous studies allows for a clearer performance comparison

__Weak points__
 - The extension to temporal networks seemed glossed over.
 - The authors refer to the parameter $r$ as the “rank parameter.” Is this related to the rank of the matrix? Is it completely defined independently of the other inputs as just the number of eigenvalues to minimize?

__Minor__
 - A table of parameters and variable descriptions would be a helpful reference
 - Do the black stripes in some colors of the bar graphs denote anything?
 - Legend missing for figure 2b

---

### Official Review · Reviewer_7pj4 · 2022-06-26
**Review for Optimal Intervention on Weighted Networks via Edge Centrality**

**Rating:** 5
**Confidence:** 4

**Review:**

The authors address the problem of diffusion control, where they consider how to minimize infections over a network by reducing the network’s edge weights, subject to a budget constraint. First, they extend the edge centrality concept from Tong et al (2012) to show that edge centrality is equal to the gradient of the largest eigenvalue of W^TW (where W is a weighted network) and that this concept generalizes to the r largest eigenvalues. Then, they introduce their algorithm, Frank-Wolfe-EC, that minimizes the sum of the r largest eigenvalues (as an approximation of infections) by iteratively applying a greedy selection of edges with the highest edge centrality scores, then recomputing the scores. This is an application of the Frank-Wolfe algorithm, and they show that their greedy selection of edges corresponds directly to the iterative step in Frank-Wolfe (which computes a descent direction for the output matrix, M). They show that their algorithm converges to the global optimum and that each iteration requires nearly-linear runtime in number of edges.

They conduct empirical studies with mobility networks that were used to model COVID-19 (Chang et al, 2021) and other weighted networks. They show that their strategy substantially reduces the number of infections resulting from SEIR spreading processes, compared to baselines. They also demonstrate through ablations the improvements gained from 1) minimizing the sum of the r largest eigenvalues, instead of only the top eigenvalue, as done in Tong et al, 2) iteratively applying greedy selection, instead of only selecting edges once.

Strengths
+ Novel algorithm for diffusion control through edge reduction
+ Interesting connections made to prior literature (edge centrality, Frank-Wolfe)
+ Useful theoretical guarantees of algorithm
+ Strong empirical results with helpful ablation experiments

Weaknesses
- Practical application not entirely clear. How would it be possible to target specific edges in a mobility network?
- Approach primarily extends existing theory and applies Frank-Wolfe, although the connections made are interesting

---

### Official Review · Reviewer_vFov · 2022-06-27
**An algorithm to reduce infections on weighted networks presented in a meticulous manuscript**

**Rating:** 4
**Confidence:** 4

**Review:**

In the wake of the ongoing pandemic, this manuscript presents an algorithm to reduce the number of infections in an epidemic spreading process using a weighted network $W$ and under the constraint of an edge-weight reduction budget. The approach consists of minimizing the largest $r$ eigenvalues of $W^TW$.
They present their approach with a strong sense of detail and show improvements over the state-of-the-art (less infected nodes and smaller largest singular value) on different datasets and with different budgets.

Pros:

- This manuscript includes proofs, a study of the complexity of the algorithm, a well-presented state-of-the-art as well as detailed explanation of the added values of the manuscript.
- A GitHub repository with the codes to allow replication of the analysis is available.

Major comments:

- It is unclear how the rank $r$ has been chosen when conducting the experiments. In the same way, it is unclear how the range of learning rates has been chosen in practice. The code on GitHub suggests (array alphas) a grid search between 1e3 and 0.9. More information regarding this choice and the impact of the learning rate range on the convergence/the performance would be appreciable.

Minor comments:

- In figure 6, I suggest adding the value of the slope.

Some typing errors in the manuscript:

- “The above problem corresponds to $\textbf{an}$ minimization problem as follow”
- Missing a 2 in equation (5) (derivation of the norm).
- Missing a $\textbf{F}$ below one of the norm in the proof of theorem 3: $||M_{1}\widetilde{U}||\_{F}^{2}+ ||M_{2}\widetilde{V}||\_{\textbf{F}}^{2}$
- “Lastly, the $\textbf{diamester}$ of the constraint set is at most”
- Conclusion: “More broadly, $\textbf{W}$e hope […]” Capital W.